# The Effect of Basalt Aggregates and Mineral Admixtures on the Mechanical Properties of Concrete Exposed to Sulphate Attacks

**DOI:** 10.3390/ma15041581

**Published:** 2022-02-20

**Authors:** Abdulhalim Karasin, Marijana Hadzima-Nyarko, Ercan Işık, Murat Doğruyol, Ibrahim Baran Karasin, Sławomir Czarnecki

**Affiliations:** 1Department of Civil Engineering, Dicle University, 21100 Diyarbakır, Turkey; karasin@dicle.edu.tr (A.K.); baran.karasin@dicle.edu.tr (I.B.K.); 2Department of Civil Engineering, Josip Juraj Strossmayer University of Osijek, Vladimira Preloga 3, 31000 Osijek, Croatia; 3Department of Civil Engineering, Bitlis Eren University, 13100 Bitlis, Turkey; eisik@beu.edu.tr; 4Department of Civil Engineering, Siirt University, 56100 Siirt, Turkey; dogruyol@siirt.edu.tr; 5Department of Materials Engineering and Construction Processes, Wroclaw University of Science and Technology, 50-370 Wroclaw, Poland; slawomir.czarnecki@pwr.edu.pl

**Keywords:** basalt aggregates, mineral additives, silica fume, fly ash, sulphate attack

## Abstract

In this study, basalt, which is common around Diyarbakır province (Turkey), is used as concrete aggregate, waste materials as mineral additives and Portland cement as binding material to prepare concrete mixes. This paper aims to determine the proper admixture levels and usability of Diyarbakır basalt in concrete mixtures based on mechanical, physical and chemical tests. Thus, in order to determine the strength and durability performance of concrete mixtures with Diyarbakır basalt as aggregate, 72 sample cubes of 150 mm were prepared in three groups: mineral-free admixture (MFA), 10% of cement amount substituted for silica fume (SFS) and 20% for fly ash (FAS) as waste material. The samples were exposed to water curing and 100g/L sulphate solution to determine the loss in weight of the concrete cubes and compressive strength was examined at the end of 7, 28 and 360 days of the specimens. Analysis of the microstructure and cracks that influence durability, were also performed to determine effects of sulphate attacks alkali-silica reactions on the specimens using scanning electron microscopy (SEM). A loss in weight of the concrete cubes and compressive strength was distinctly evident at the end of 56 and 90 days in both acids.

## 1. Introduction

Approximately 70–75% of concrete volume is composed of aggregates. Among the commonly preferred concrete aggregates are limestone and dolomite. However, given that concrete is the most commonly used building material the world over, it is often necessary to make use of alternative rocks [1]. Basalt is the most common type of igneous rock on the Earth’s surface. Having been used successfully in old structures for long years, basalt, a stiff and durable material abundant on arth with a high strength, may be preferred as an alternative of concrete aggregate [2]. In order for basalt to be used as aggregate, it should be resistant to granulometry, abrasion and frost and should not cause chemical degradation in concrete. In addition, it should not contain noxious substances such as clay and silt which can potentially affect concrete and bond strength [3].

Some studies were done on basalt aggregate concretes in the middle of 20th century, the results of which revealed that basalt rocks were superior compared to other rocks. The thermal expansion coefficient, conduction and conduction velocity of basalt are low. Neither temperature rises nor crack formation was observed during the cooling [4,5,6]. During the compression tests, it was determined that basalt aggregate concretes had higher compression strength, that the compression strength of basalt rock was higher than that of limestone and that water absorption and voids had low values [7]. Some studies dealing with basalt have concluded that basaltic aggregates increase the quality of concrete [8,9].

Basalt is formed mainly from feldspar under the natural stones classification and is dark grey and black in color in the nature, being a stiff and durable volcanic rock. It is commonly found in various properties in Diyarbakır province and its neighborhood, the East and Central Anatolia and Thrace regions in Turkey [10,11]. Basalt is more cost-effective in comparison to other aggregates in as much as it is abundant in the region with lower processing costs due to the advances in today’s technology.

Supplementary cementitious materials (SCMs) can reduce the amount of cement in cementitious mixes without significantly changing their mechanical properties. Silica fume (SF), comprised primarily of microscale size SiO_2_ particles has been widely used to produce high strength concrete. SF as a highly pozzolanic material that can participate in the pozzolanic reactions occurring during the very early stages of cement hydration. In addition, because it has very small particles, SF could also act as a filler in the pores and voids in concrete to improve the permeability. Moreover, SF particles contribute to the precipitation of hydration products that accelerate cement hydration leading to hardened concrete with a refined pore structure [12,13,14]. On the other hand, the research concerning the impact of fly ash (FA) as one of the most commonly used SCMs in cementitious mixes and curing conditions on fracture toughness and the cracking mechanism in concrete has indicated that it is possible to make sustainable green concrete with the addition of fly ash with high mechanical properties. On the other hand, fly ash decreases the early age strength of concrete and that the curing conditions have a significant impact on the properties of concrete with FA [15,16,17]. Moreover, it is noted that FA which has become a useful material that is used in the production of cementitious mixtures is no longer considered a waste [18].

The International Union of Geological Sciences (IUGS) has reported that the rock whose SiO_2_ amount and total of Na_2_O + K_2_O alkaline do not exceed 45–52% and 5% respectively according to the systematic of igneous rocks basic features and that its use in concrete does not lead to ASR [19]. In accelerated mortar bar tests conducted in accordance with ASTM C 1260 standard, it was identified that basalt is the most nonaggressive aggregate that forms ASR compared to sedimentary rocks such as opal, chert and chalcedony [20].

In this paper, to heighten the strength and durability of concrete, Diyarbakır Karacadağ crushed basalt and waste mineral admixtures (fly ash, silica fume), by-products of various industries which resemble cement, were used as concrete aggregate. Proper admixture levels in concrete mix were tested to determine the mechanical, physical and chemical properties for various conditions. Moreover, the effects of hazardous environments such as sulphate attacks on the specimens discussed. This paper is prominent as it aims to determine and usability of Diyarbakır basalt as an alternative aggregate material obtained from the Tigris river stream bed. Utilization of Diyarbakir basalt in concrete industry will not only avoid the use of aggregates obtained from streamside and thereby avert degradation of stream beds but also will allow reclamation of lands from which the basalt is removed to provide farmland.

### 1.1. Physical, Chemical and Mechanic Properties of Diyarbakir Basalt

Diyarbakır and Karacadağ mountain basalt, found locally in deposits as thick as 150 m cover an area of 10,000 km^2^ in Southeast Anatolia, Turkey and occur approximately a half meter below the soil cover or as a surface cover. The creeping basalt flow from Karacadağ volcano that formed these basalt rocks is common, most notably in the Diyarbakır wide basalt plateaus that are found across 120 to 130 km.

Diyarbakir Karacadağ basalt rock is usually dark grey or black in colour. Some other basalt rocks may weather to lighter colours. They may also weather to a brown colour following oxidation of ferrous minerals [21]. Diyarbakır basalt has two types, designated as porous and non-porous basalt, with densities that vary from region to region. It has values of 2.66–2.79 g/cm^3^ in the west, 2.91–2.93 g/cm^3^ in the north and 2.45–2.47 g/cm^3^ in the east, respectively. Mean uniaxial strength of Diyarbakır basalt rock (five samples) with a diameter of 42 mm and height of 102 mm is 51.76 MPa for the porous form and 89.10 MPa for the non-porous one [22]. A comparison of the physical and mechanic properties of basalt with the limestone, yet another rock, is provided in Table 1 [23].

Other properties of Diyarbakır-Karacadağ basalt such as high abrasion strength, low thermal conductivity and acid and frost resistance have diversified the areas of its use and led to an increase in the number of studies related to the material being performed nowadays. Kahveci and Kadayifçi [24] have reported basic element chemical analyses of samples taken from four points close to the Karacadağ summit, results obtained through X-ray florescence spectrometry (XRF) at Earthquake Research Institute of University of Tokyo as well as chemical analysis results of testing basalt obtained from Diyarbakır-Elazığ region at Dicle University labs. The chemical analysis values of Diyarbakır basalt are given in Table 2.

### 1.2. Alkali-Silica Reaction (ASR) in Concrete

Alkali silica reaction (ASR) is a form of concrete degradation occurring as a result of available reactive SiO_2_ in aggregates used in concrete admixtures leading to a reaction with Na_2_O and K_2_O (alkali) components that form the cement [25]. ASR-based deterioration in concrete was first observed in the 1940s and in 1995 in Turkey around the İzmir region in some highways and bridges nearby [26]. Reactions caused by ASR occur in two steps [27]:H_0.38_ SiO_2.19_ + 0.38NaOH → Na_0.38_ SiO_2.19_ + 0.38H_2_O(1)
Reactive Silica + Alkali → Alkali Silica Gels + Water(2)
Na_0.38_ SiO _2.19_ + 1.62NaOH + 0.38H_2_O → 2Na^+^ H_2_SiO^2−^_4_(3)
Alkali Silica Gels + Alkali + Moisture → Alkaline Silicate Hydrate Gel (Expansion)(4)

The product of alkali silica reaction is a highly hydrophilic alkali silica gel [28]. As alkali silica gel absorbs moisture, the concrete expands up to 2–3% in volume. Having expanded due to moisture, alkali silica gel forms fractures and cracks in and on the surface of the concrete in unreinforced mass concretes, coating concretes as a result of a decrease in tensile strength, leading to loss in durability [29,30]. Typical ASR-induced cracks observed at Izmir Adnan Menderes airport shown in Figure 1 [31].

Considering the chemical data in Table 2, all samples taken as Diyarbakır basalt have basic characteristics according to the chemical classification of volcanic rocks using a total alkali-silica TAS diagram that represent the basalt rock formation in central parts of Diyarbakır [25]. Considering the chemical results of the Diyarbakır basalt samples withgiven in Table 2, it is noted that according to [25] it is basic and according to [32] it is not a potential reactive aggregate to ASR as its SiO_2_ content is less than 51% [25,32]. The study identified cements whose Na_2_O amount is 0.50% as low alkali and cements with 1.04% amount as high alkali. Where highly reactive rocks to ASR and low alkali cements are used, the ASR impact has been identified to be reduced. Chemical analysis of the cement applied during the test is given in Table 3 and it is understood that it belongs to the low alkali category according to [32].

It was observed that ASR products generally are located in voids, in-between cracks of aggregates and aggregate-cement paste. Having different morphological structures, massive ASR products can be clearly seen in Figure 2a,b with images zoomed by 35× and 650× respectively [33].

The effects of alkali-silica reactions were investigated by scanning electron microscopy (SEM) for mortar samples of granite aggregates and CEM I 42.5R Portland cement [34]. According to the results obtained, granites as concrete aggregate with 68% SiO_2_ content cause cracks that create a volumetric expansion due to alkali-silica reactions, as shown in Figure 3.

The alkali silica reaction (ASR) effects on mortars containing various proportions of silica fume (SF) and fly ash (FA) as replacement of cement were investigated with respect to the ASTM C 1260-07 standards [35]. It is noted that the least strength loss in the mortars exposed to ASR occurred in those samples which had 20% SF substitution that provided 18% better flexural strength and compressive strength compared to the mortars with FA.

## 2. Materials and Methods

In this paper, as admixture materials fly ash (FA) from the Kahramanmaraş Afşin Elbistan Thermal Reactor (Afşin, Turkey), was taken. Silica fume (SF) was from the İstanbul Aryum Company (Istanbul, Turkey) and cement from the Limak Ergani Cement Plant (Limak Holdings Inc., Ankara, Turkey) of Portland 42.5 cement category that is similar to ASTM C type have been utilized, as tabulated in Table 3.

Analysis results of mineral admixtures were conducted at the Limak Ergani Cement Plant. Determinations of specific surface area of the materials were made via a Blaine test using Blaine test equipment in compliance with TS EN 196:6, ASTM C 204 standards, and an approximate value of the specific surface area of silica fume was obtained using the nitrogen adsorption method. Physical and chemical analysis results of the materials applied during the test are shown comparatively in Table 3.

Table 3 indicates that the fly ash belongs to the Class F fly ashes in which, according to the TS639 and ASTMC618 standards, the percentage of SiO_2_ + Al_2_O_3_ + Fe_2_O_3_ is more than 70. Class F fly ashes are pozzolanic and do show binding properties on their own. Silica fume, used in concrete as admixture, has a SiO_2_ content of more than 85%. Oxides like Al_2_O_3_, SO_3_, MgO, Na_2_O and K_2_O are found in amounts of less than 1% [36]. On the other hand, crushed basalt rock, which was supplied from Karacadağ in the Diyarbakır-Elazığ region, was used as aggregate in the concrete mix.

The sieve analysis according to TS 1225 shown in Figure 4 represents the physical properties of aggregates formed from fine aggregate (0–8 mm) and two groups of coarse aggregate (8–16 mm) and (16–32 mm). Results of some experiments carried out in accordance with adequate codes to define the specific gravity, abrasion loss and water absorptions of the basalt aggregates are given in Table 4.

The concrete samples were prepared in 72 cubes of 150 mm divided into three groups in which mineral free admixture (MFA) was available and 10% of cement amount was substituted for silica fume (SFS) and 20% for fly ash (FAS). The specimens were exposed to 100g/L sulphate solution and crushed at the end of days 7, 28 and 360 to determine hydropathy and durability. According to ASTMC1012, 50g/L magnesium sulphate solution used. However, in the previous study it was observed that the relevant solution failed to exhibit differences in relation to sulphate durability of prepared admixtures [37].

Fifteen cm cubic concrete samples prepared for the test were designed in compliance with TS 802 (2009) standards and C30 concrete class. A superplasticizer whose specific weight was 1.15 g/cm^3^ was used as chemical admixture. In modified concretes, cement was replaced by admixtures FA and FS by their weight, the content of 1 m^3^ concrete values are given in Table 5.

According to the classification of samples calculated with the slump amounts via slump test provided in TS EN 201-1 given in Table 5, the sample with no mineral admixture from which fly ash substitute concrete was obtained goes into the S3 class and that of silica fume substitute belongs to the S2 class.

## 3. Results and Discussion

### 3.1. Concrete Compressive Values

The three test groups were constituted with the replacement percentages of 0% (the plain concrete-MFA), 10% (SFS-10) and 20% (SFA-20) specimens were cured for 7, 28 and 360 days under laboratory conditions. Concrete compressive strength values, mean value and strength were calculated by crushing three samples, each of which had different designs, through pressure tests run with a hydraulic system on which the load applied was set at 0.6 N/mm^2^/sec. Comparative compression values are given in Figure 5 and Figure 6.

Regarding the variation of the strength rates between hydropathy and sulphate solution considered for the short term (7-days) MFA did not affect these parameters, but the FAS samples were very affected, showing about a 21% decrease in strength. On the other hand, comparing the compression strength values for days 7, 28 and 360 within hydropathy and sulphate solution, silica fume substitute concrete specimens demonstrated the highest performance. Whereas fly ash substitute concrete demonstrated the lowest strength at an early age, it approached the free admixture concrete strength value by day 28, and its strength value at an advanced age measured on day 360 turned out to be higher than that of a admixture-free concrete. The change in compression strength values of concrete samples soaked in hydropathy and sulphate solution on day 360 and day 28 are given in percentages in Figure 7.

Figure 7 implies that at the end of 360 days, the compression increase of samples in hydropathy continued. There was a decrease in compression by 7.72% and 1.46%, respectively in free admixture and silica fume substitute samples, while an increase by 6.47% was observed only for fly ash substitute samples. This situation proves the corrosive impact of the sulphate on concrete. It can be concluded that the positive impact of fly ash is more successful than the negative impact of sulphate attacks.

### 3.2. Mass Loss of Concrete

The mass loss ratios of samples kept in water and sulphate solution for 7, 28 and 360 days are given in Figure 8 and Figure 9.

Figure 8 presents the highest mass loss at the end of 360 days measured for MFA samples (4.52%). However, during the same period a 1.63% loss was measured for SFS and a mass increase of close to 1% was measured for the FAS samples. On the other hand, in all cases at the end of 7, 28 and 360 days in hydropathy the MFA specimens presented the highest mass loss of concrete compared to the others. On the contrary, the FAS specimens presented the best performance for the same cases. The FAS specimens gains in weight at 7 days cause a strength loss, but for the long-term (360 days) stage they display a rapid increase in strength as indicated in Figure 5.

As stated in Figure 9 for the sulphate solution case, the highest mass loss was measured for the MFA samples (4.79%). On the other hand, no significant change was measured for the FAS and SFS samples (less than 1.23%). It is noted that the MFA samples behave as in the hydropathy case. However, under both curing conditions the replaced admixtures provided a lower concrete mass loss.

### 3.3. SEM Observations

In order to investigate effects of mineral admixtures on the microstructure of various concrete bar samples of plain concrete and mineral additive replacement of CEM I 42.5 Portland cement observations by scanning electronic microscopy (SEM) were performed The various bar specimens of Diyarbakır basalt aggregates with %50 SİO_2_ content cured in sulphate solution and hydropath for 90 days are shown in Figure 10.

The effects of alkali-silica reactions were observed by SEM in the hydropath samples after curing for 90 days, as shown in Figure 11, Figure 12 and Figure 13. It is noted that for all of the cases no cracks at the bar samples that create volumetric expansion due to alkali-silica reactions were observed.

For the samples cured for 90 days in sulphate solution, the effects of alkali-silica reactions were observed by SEM as shown in Figure 14, Figure 15 and Figure 16.

In the SEM images shown in Figure 14, the control bar samples without mineral additives cured for 90 days in sulphate solution surface display some deteriorations and cracks that are not observed in the hydropath cure. These deteriorations and cracks are thought to be caused by sulphate attacks. However, it is noted that no surface deterioration due to the sulphate attacks was observed and only partially thinner cracks were formed in the SD substituted samples shown in Figure 15. On the other hand, since there is no surface deterioration with capillary size cracks it is possible to conclude that the FA-substituted samples shown in Figure 16 are less affected by sulphate attacks.

## 4. Conclusions

Utilization of Diyarbakir basalt in the concrete industry as a locally available raw material with high strength capacity will not only prevent the use of aggregates obtained from riversides, and thereby avert the degradation of river beds, but also will allow reclamation of lands from which the basalt is mined to provide farmland. On the other hand, the use of waste minerals such as fly ash with appropriate proportions of cement replacement in concrete mix saves both the energy consumed for cement production and carbon emissions, addressing an important environmental issue.

As it is noted the samples produced with Diyarbakır basalt aggregate did not show any volumetric cracks after 90 days of curing and were not reactive to ASR as the microstructure seen in SEM images did not change and products related to alkali-silica reaction were not observed. The chemical analyses of all five samples revealed that there is no objection in terms of ASR for the use of Diyarbakir basalt as concrete aggregate. Additionally, the mineral admixtures such as FA can improve the microstructure as well as the compressive strength reducing the mass loss of concrete. The following conclusions are derived to summarise our results in view of the results and discussion presented in this research work:The compression strength results of 28-day-old concrete with basalt aggregates (43–50 MPa) indicate that samples designed as C30 (30 MPa) class exhibit much better performance than those expected with respect to the standards for both hydropathy and sulphate solution curing conditions.Silica fume substitute concrete specimens achieved the highest compression strength both in hydropathy and sulphate solution at the early and long-term stages.The most economical specimens of fly ash as partial replacement of cement in concrete mix performed close to the 28-day compressive strength of the plain concrete samples.In the long-term case the FA replacement specimens provided improvements in strength for both in hydropathy and sulphate solution.In the short-term case the FA replacement specimens present weak behaviour in strength and mass loss cases for both curing conditions.In all cases of short and long-term (7, 28 and 360 days) under both curing conditions the cement-replaced admixtures (10% SFS and 20% FA) of concrete specimens provided lower concrete mass lossesf.Large cracks were observed in the MFA (plain concrete) and SFS (10% silica fume substituted) samples that were exposed to the sulphate solution for 90 days.In the SEM observations of the 20% FA substituted samples exposed to sulphate solution only capillary cracks were observed which represents an improvement in crack formation.It is clearly noted that the sulphate-resistant samples containing fly ash perform much better than the other samples in terms of durability and concrete strength over long-term periods.

## Figures and Tables

**Figure 1 materials-15-01581-f001:**
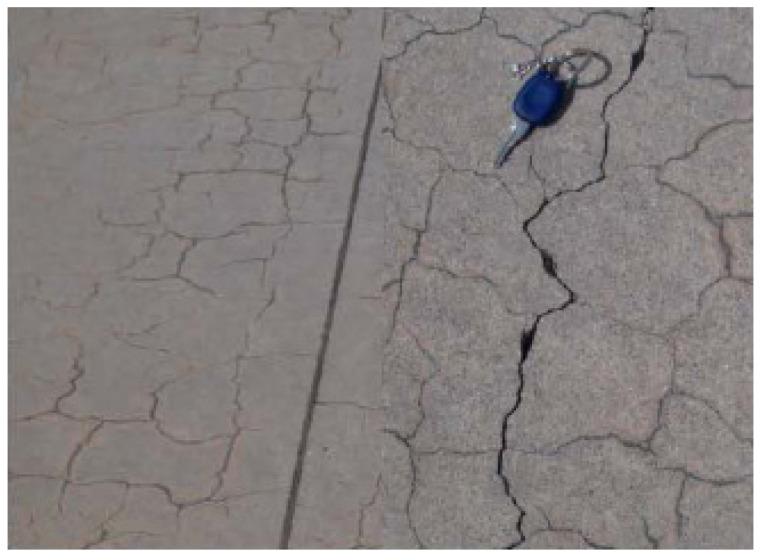
ASR impact on concrete.

**Figure 2 materials-15-01581-f002:**
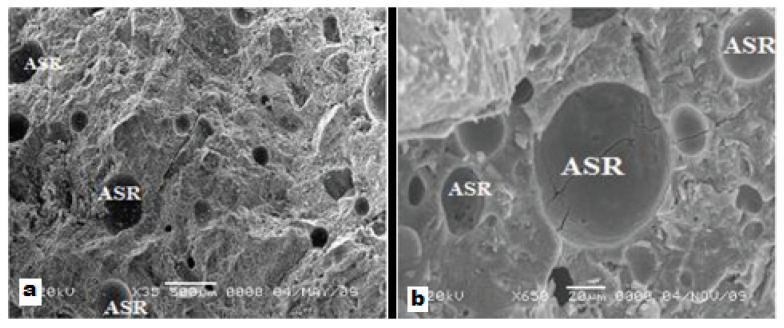
ASR products monitored via SEM, (**a**) 35×; (**b**) 650×.

**Figure 3 materials-15-01581-f003:**
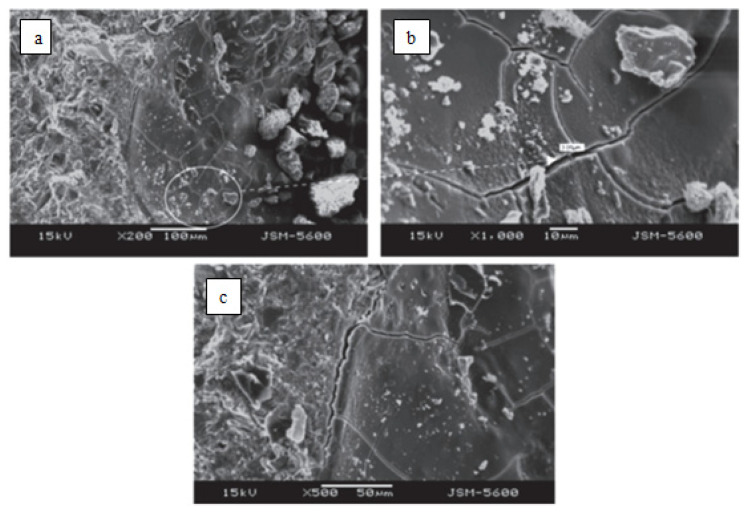
SEM images of mortar bar samples of granite: (**a**,**c**) typical expansion cracks produced in gel, (**b**) close view of same cracks [34].

**Figure 4 materials-15-01581-f004:**
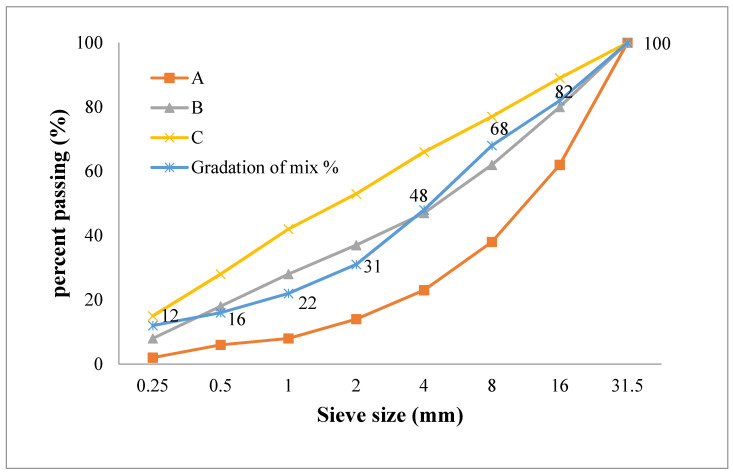
Sieve analysis of the basalt aggregate.

**Figure 5 materials-15-01581-f005:**
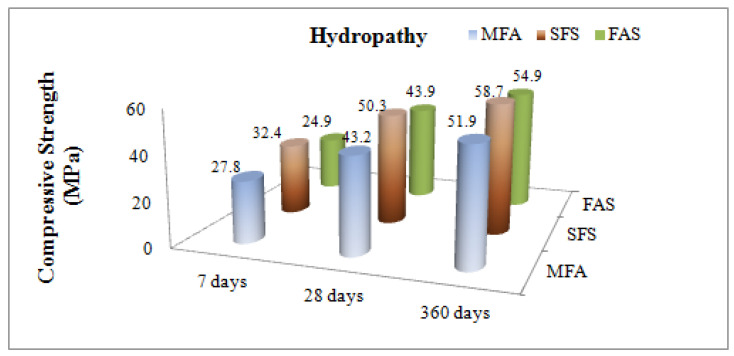
Effects of mineral admixtures on compressive strength of concrete for 7, 28 and 360 days in hydropathy.

**Figure 6 materials-15-01581-f006:**
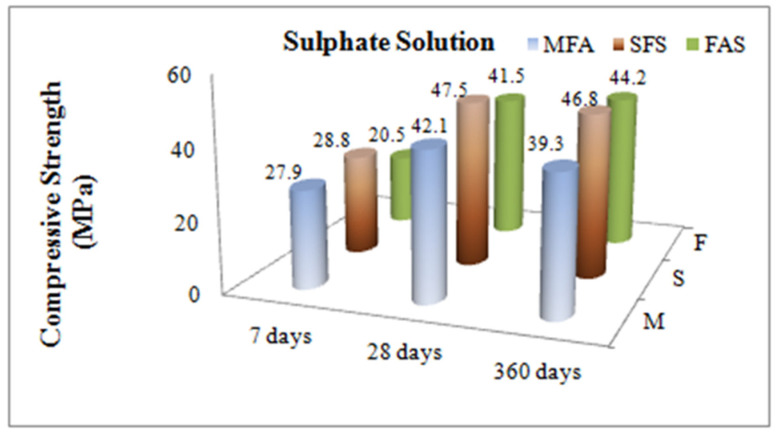
Effects of mineral admixtures on compressive strength of concrete for day 7, 28 and 360 exposed to sulphate solution.

**Figure 7 materials-15-01581-f007:**
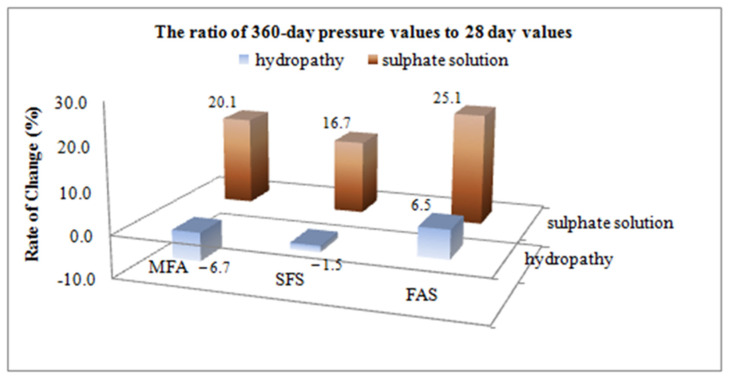
Effects of mineral admixtures on strength variation for ratio of 360 to 28 days values.

**Figure 8 materials-15-01581-f008:**
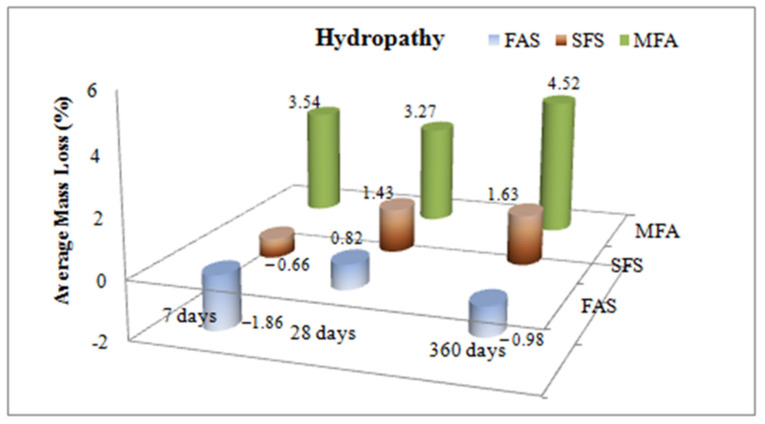
Effects of mineral admixtures on mass loss in hydropathy.

**Figure 9 materials-15-01581-f009:**
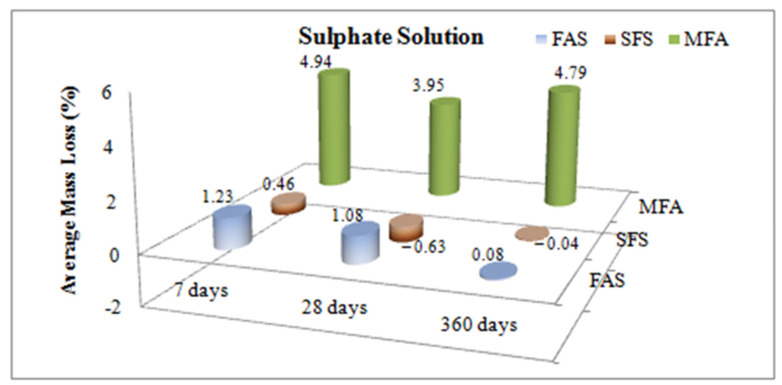
Effects of mineral admixtures on mass loss in sulphate solution.

**Figure 10 materials-15-01581-f010:**
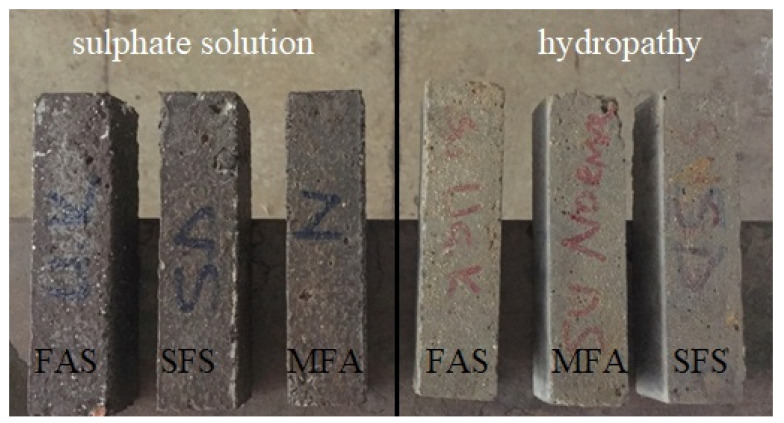
Plain concrete (MFA), silica fume (SFS) and fly ash (FAS) mineral additives substituted bar samples with curing conditions in hydropath and sulphate solution.

**Figure 11 materials-15-01581-f011:**
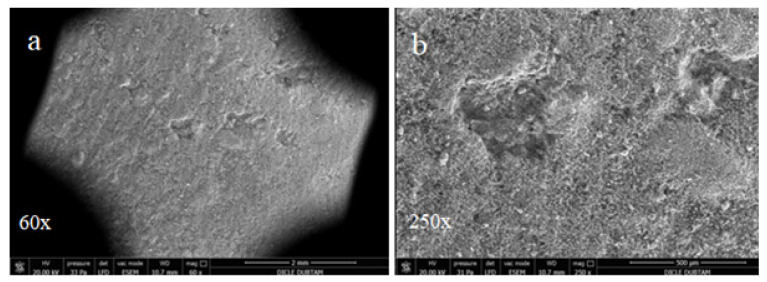
SEM images of control bar samples without replacement cured for 90 days in hydropath: (**a**) 60×; (**b**) 250×.

**Figure 12 materials-15-01581-f012:**
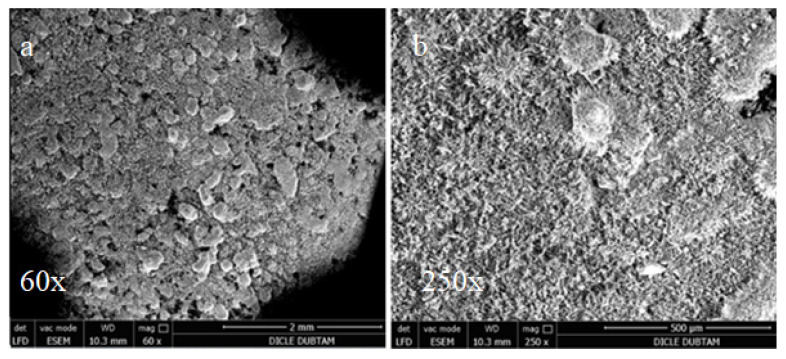
SEM images of SFS replacement bar samples cured 90 days in hydropath: (**a**) 60× (**b**) 250×.

**Figure 13 materials-15-01581-f013:**
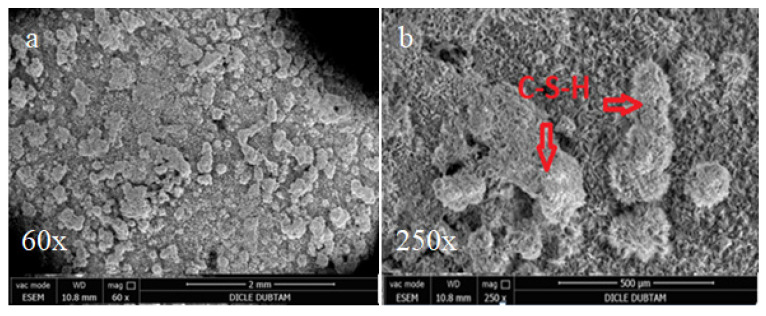
SEM images of FAS replacement bar samples cured 90 days in hydropath: (**a**) 60×; (**b**) 250×.

**Figure 14 materials-15-01581-f014:**
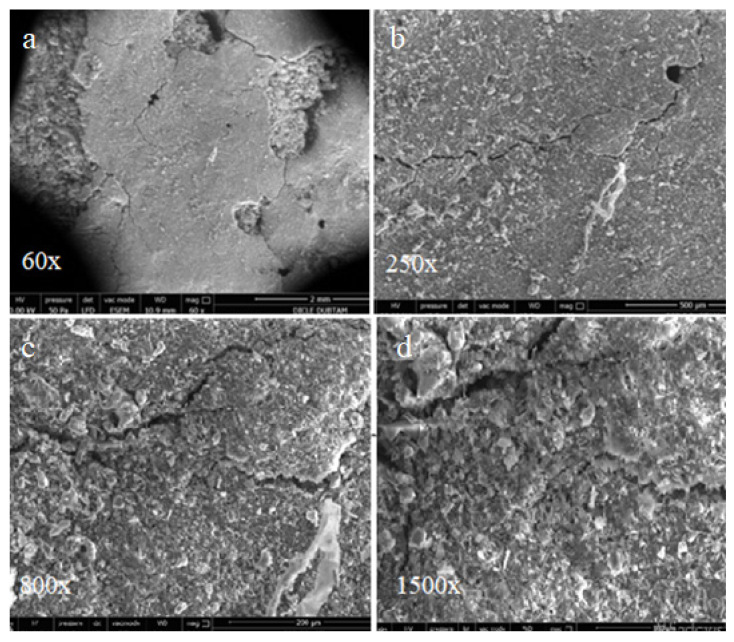
SEM images of control bar samples without replacement cured for 90 days in sulphate solution: (**a**) 60×; (**b**)250×; (**c**) 800×; (**d**) 1500×.

**Figure 15 materials-15-01581-f015:**
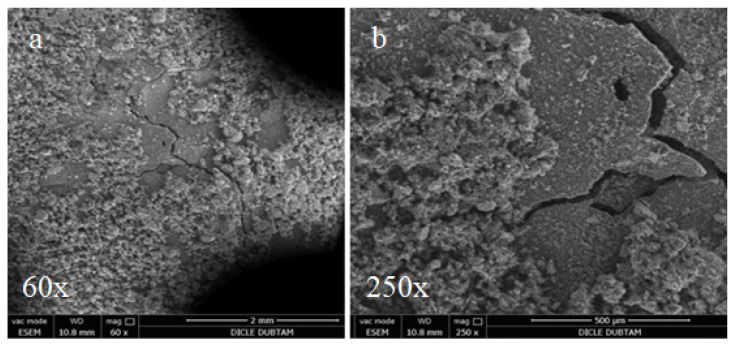
SEM images of SFS replacement bar samples cured for 90 days in sulphate solution: (**a**) 60×; (**b**) 250×.

**Figure 16 materials-15-01581-f016:**
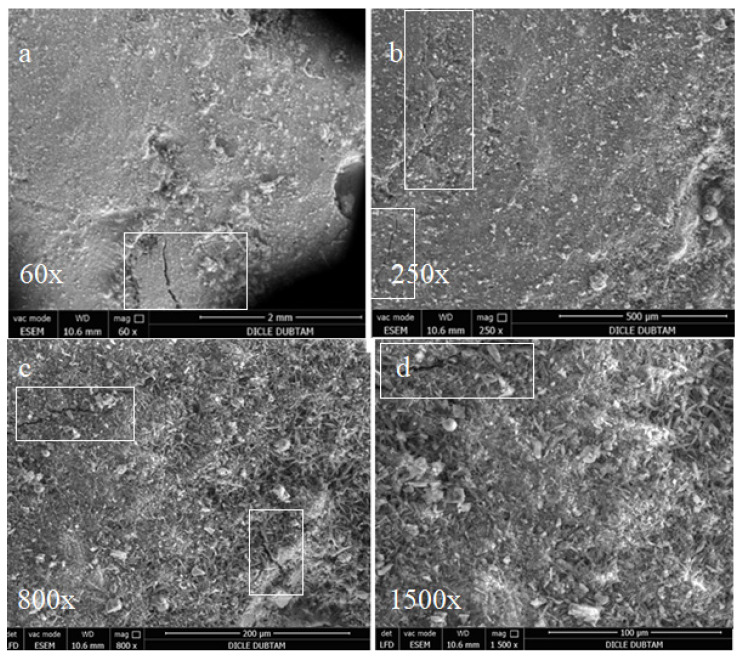
SEM images of FA replacement bar samples cured for 90 days in sulphate solution (**a**) 60×; (**b**) 250×; (**c**) 800×; (**d**) 1500×.

**Table 1 materials-15-01581-t001:** Comparison of physical and mechanic properties of basalt with the limestone.

Some Properties	Basalt	Limestone
Compressive Strength (MPa)	97.60	17.80
Flexural Strength (MPa)	16.20	4.90
Schmidt hammer (MPa)	75.00	21.00
Compressive Strength after frost deteriorated (MPa)	83.90	16.30
Modulus of Elasticity (GPa)	68.50	14.00
Poisson Ratio	0.25	0.31
Water absorption by weight (%)	0.81	8.40
Water absorption by volume (%)	2.17	19.22
Water absorption (g/cm^3^)	0.022	0.174
Capillary Abrosption (cm/sn)	1.52	1.23
Unit Weight (g/cm^3^)	2.69	2.08
Density gap amount (%)	6.80	19.22
LA Abrasion loss (%)	18.75	27.76
Freeze-thaw loss (%)	6.90	0.07
Thermal conductivity (W/m.K)	1.74	1.42

**Table 2 materials-15-01581-t002:** Chemical analyses of Diyarbakır basalt aggregate (%).

Components	Sample 1 [24]	Sample 2 [24]	Sample 3 [24]	Sample 4 [24]	Test Sample (Present Study)
SiO_2_	46.04	48.70	46.79	45.88	50.28
TiO_2_	3.18	2.62	2.94	2.86	1.00
Al_2_O_3_	13.89	14.05	14.18	13.68	22.60
Fe_2_O_3_	4.89	4.32	4.62	4.69	3.24
FeO	8.80	7.78	8.31	8.44	-
MnO	0.15	0.15	0.16	0.16	-
MgO	8.76	8.57	9.19	8.82	4.55
CaO	9.12	8.67	8.77	9.32	5.76
Na_2_O	3.64	3.23	3.18	3.78	6.00
K_2_O	1.04	1.36	1.30	1.63	1.50
P_2_O_5_	0.44	0.48	0.51	0.69	0.50

**Table 3 materials-15-01581-t003:** Physical and chemical analysis results of the materials.

Components	SF	FA	CEM I 42.5
SiO_2_, %	90.50	58.02	21.13
TiO_2_, %	0.02	1.76	-
Al_2_O_3_, %	1.86	23.31	4.98
Fe_2_O_3_, %	2.63	5.91	3.73
MgO, %	0.78	1.88	1.13
CaO, %	0.42	3.78	65.23
Na_2_O, %	0.16	0.84	0.28
K_2_O, %	0.97	1.76	0.78
SO_3_, %	0.14	0.05	3.09
Other components			
Loss on Ignition	1.68	2.46	1.29
Humidity	0.67	0.12	-
Specific Gravity (g/cm^3^)	2.22	2.20	3.15
Blaine (cm^2^/gr)	~ 2000	2172	3874

**Table 4 materials-15-01581-t004:** Some important properties of the aggregates and limit values.

	Aggregate Groups				Acceptance Value	Standard
Properties	Coarse Aggregate (16–32)	Coarse Aggregate (8–16)	Fine Aggregate (0–8)	Aggregate Mix		
Specific Gravity	2.797	2.792	2.733	2.76	<2.70	ASTM C-127
Water absorption, %	1.082	1.317	2.712		<2.0	ASTM C-127
Los Angeles Abrasion Loss, %	30.5				<50	ASTMC 131-03 TS 706

**Table 5 materials-15-01581-t005:** 1 m^3^ concrete admixture values and fresh concrete contents.

Materials	MFA	20% FAS	10% SFS
Portland Cement (kg/m^3^)	350	280	315
Water (kg/m^3^)	150	150	150
Coarse aggregate (16–32) (kg/m^3^)	496	464	480
Coarse aggregate (8–16) (kg/m^3^)	413	386	399
Fine aggregate (0–8) (kg/m^3^)	1131	1058	1094
Fly Ash(kg/m^3^)	0	70	0
Silica Fume (kg/m^3^)	0	0	35
Chemical addictive (Superplasticizer) (kg/m^3^)	3.5	3.5	3.5
Unit weight (kg/m^3^)	2543	2411	2477
Slump (cm)	14	15	6
Water /Binding	0.43		

## Data Availability

Data sharing not applicable.

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
