# Peer review of "The Effect of Basalt Aggregates and Mineral Admixtures on the Mechanical Properties of Concrete Exposed to Sulphate Attacks"

_materials, 2022, doi:10.3390/ma15041581_

Round 1

Reviewer 1 Report

An interesting work, however should be improved in a number of aspects before final acceptance. 

1. Introduction section should be revised removing the redundant information about the aggregate. The focus of the paper should be made on the mechanical and micro- structure behavior of concrete.

2. Figures 1 and 2 are redundant and can be removed. 

3. Provide reference for the properties specified in Table 1.

4. Table 2: Please elaborate how the properties of the individual oxides are obtained.

5. The reviewer feels that figure 3 is redundant and ca be removed. The paper mostly presents a lot of available information. 

6. Table 4 can be presented in the form of a graph for better understanding. 

7. Explanation of results in section are to  e significantly improved.  At present, it seems like the obtained values are just presented.

Reviewer 2 Report

This is a well written paper. However, only moderate corrections are required as per the comments attached.

Reviewer 3 Report

Dear Authors,

Thank you for manuscript, here will be following comments:

  1. Abstract has to elaborated more, plea indicate novelty and main results outcome.
  2. When resubmit next time your manuscript make sure the Line Numbers is continuous.
  3. Intro: - RQs + goals + innovative aspects + …??? References to similar work by other research?
  4. Equations (1) and (2) are prtscr??!
  5. Sections 1.1, 1.2 -> belongs to Materials Chapter
  6. 2.3 Section should be Methodology or as good paragraph in Intro? Work out this section better, indicate only necessary details related to the subject of the research of your paper.
  7. PSD od the materials?
  8. there is no Methodology chapter as such! Please provide a detailed info on your test procedures and mixing. Where is you REFERENCE mix? How do you compare results?! Please provide in table REF mix design and calculus of all components to be sure that you have volume of 1.0 in 1m3.
  9. Charts quality is poor, improve!
  10. SEM images – please provide a better description.
  11. Conclusion have to be elaborated and clearly state the outcome (bullet form – short sentences with clear results) and novelty of research.

Round 2

Reviewer 1 Report

Comments were addressed and the quality has been improved when compared to the original manuscript. Hence, the paper can be accepted for publication.

Reviewer 3 Report

Dear Authors, no further remarks